# Tunable Spin and Orbital Edelstein Effect at (111) LaAlO_3_/SrTiO_3_ Interface

**DOI:** 10.3390/nano12142494

**Published:** 2022-07-20

**Authors:** Mattia Trama, Vittorio Cataudella, Carmine Antonio Perroni, Francesco Romeo, Roberta Citro

**Affiliations:** 1Physics Department “E.R. Caianiello”, Universitá degli Studi di Salerno, I-84084 Fisciano, Italy; fromeo@unisa.it (F.R.); rocitro@unisa.it (R.C.); 2INFN—Sezione di Napoli, Complesso Universitá Monte S. Angelo, I-80126 Napoli, Italy; 3Physics Department “Ettore Pancini”, Universitá degli Studi di Napoli “Federico II”, Complesso Universitá Monte S. Angelo, I-80126 Napoli, Italy; cataudella@na.infn.it (V.C.); carmine.perroni@unina.it (C.A.P.); 4CNR-SPIN Napoli Unit, Complesso Universitá Monte S. Angelo, I-80126 Napoli, Italy

**Keywords:** Edelstein effect, spin-orbit, orbital magnetization, spintronics, orbitronics, oxide heterostructures

## Abstract

Converting charge current into spin current is one of the main mechanisms exploited in spintronics. One prominent example is the Edelstein effect, namely, the generation of a magnetization in response to an external electric field, which can be realized in systems with lack of inversion symmetry. If a system has electrons with an orbital angular momentum character, an orbital magnetization can be generated by the applied electric field, giving rise to the so-called orbital Edelstein effect. Oxide heterostructures are the ideal platform for these effects due to the strong spin–orbit coupling and the lack of inversion symmetries. Beyond a gate-tunable spin Edelstein effect, we predict an orbital Edelstein effect an order of magnitude larger then the spin one at the (111) LaAlO${}_{3}$/SrTiO${}_{3}$ interface for very low and high fillings. We model the material as a bilayer of $t_{2g}$ orbitals using a tight-binding approach, whereas transport properties are obtained in the Boltzmann approach. We give an effective model at low filling, which explains the non-trivial behaviour of the Edelstein response, showing that the hybridization between the electronic bands crucially impacts the Edelstein susceptibility.

## 1. Introduction

Spintronics is an emergent field that exploits the intrinsic spin of the electrons, in addition to its charge. The goal is to produce devices that combine information storage, sensing, and processing in a single platform. In view of their characteristics, these devices could, in principle, overcome the performance of standard electronic devices in terms of data processing speed and consumption [1]. A possibility for spin manipulation is the injection of spin current from ferromagnets to semiconductors, which is, however, inefficient [2].

The best option is provided by the spin-to-charge interconversion, which allows for generating spin current directly inside the materials. In non-magnetic systems, this can be realized, either by the spin Hall effect or the Edelstein effect (EE). The former is the creation of a transverse spin current in response to a charge current [3], whereas the latter is the spin accumulation in response to an applied electric field [4,5]. This spin accumulation can be explained as follows: an electric field $\vec{E}$ shifts the Fermi surfaces of the non-degenerate Kramers doublets, leading to an imbalance of spin and, consequently, a magnetization occurs.

A crucial role in obtaining the EE is the lack of inversion symmetry, which causes a Rashba spin–orbit coupling (SOC), locking the spin with the quasi-momentum of the electrons in a crystal. Oxide heterostructures are therefore the perfect environments for such a coupling. The interface between the two insulating materials generates a quantum well for the electrons, forming a quasi-two dimensional electronic gas (2DEG), which naturally lays in a system with lack of inversion symmetry [6]. Moreover, in these oxides the atomic SOC is typically stronger than in semiconductor interfaces due to the *d* orbitals of the atom involved in the crystal structure [7]. Therefore, (001) SrTiO${}_{3}$ (STO)-based heterostructures exhibit many non-trivial phenomena based on spin–orbital motion, such as tunable SOC [8], generation and control of spin and orbital textures [9], coexistence of superconductivity and 2D magnetism [7,10], and topological properties both in normal and superconducting states [11,12,13,14,15,16,17,18,19,20]. Even if the inverse Edelstein effect, namely, the generation of a charge current in response to a spin current, has been studied more extensively [21], only recently has the EE been taken into account in this system [22,23]. The results are promising, not only due to the presence of the canonical EE, but also because of the presence of the so-called orbital Edelstein effect (OEE) [24], making this interface appealing for the field of orbitronics [25]. Because the electrons of 2DEG have a *d* orbital character, an orbital magnetization occurs in response to an electric field.

The promising results obtained so far with (001) interfaces further motivates the interest into interfaces along other crystallographic directions. The (111) direction has been recently proven to be particularly promising, due to the hexagonal lattice of these structures, which is responsible for many non-trivial phenomena [26,27,28]. The (111) LaAlO${}_{3}$/SrTiO${}_{3}$ (LAO/STO) interface has been intensively studied [29,30,31,32,33,34]. However, there are no predictions or experimental evidence on EE or OEE in this system, even though both the material and the direction are particularly interesting. In this system, the strong orbital intermixing and the peculiar spin and orbital textures [35,36,37] suggest the possibility of establishing an orbital magnetization and could be of practical interest for the realization of spintronics and orbitronics devices.

Therefore, in this work we theoretically predict the existence of the EE and OEE in the (111) LAO/STO interface, characterizing its properties. We model the material via a bilayer of Ti atoms using three orbital degrees of freedom treated by the tight binding (TB) approach, whereas the transport properties are modeled within the relaxation time approximation of the Boltzmann approach. We predict two different behaviours of the electrical response: a gate-tunable spin EE and an OEE an order of magnitude higher than the spin one, which cannot be explained in a common simplified Rashba model. We show that they emerge from the combined effect of the non-trivial Rashba SOC and the multi-orbital character of the electronic band structure.

## 2. Methods

The electronic band structure of the LAO/STO interface can be obtained in terms of the $t_{2g}$ orbitals of the Ti atoms in STO [38]. In order to take into account the electronic confinement, we use an accurate TB model, described in Ref. [37], of two layers of Ti atoms projected in the (111) direction, resulting in a honeycomb lattice, as shown in Figure 1. The Hamiltonian we take is
(1)$$H=H_{\mathrm{TB}}\left(t_{D},t_{I}\right)+H_{\mathrm{SOC}}\left(\lambda \right)+H_{\mathrm{TRI}}\left(\Delta \right)+H_{v}\left(v\right),$$
where $H_{\mathrm{TB}}$ contains the direct and indirect first neighbour hopping terms, whose amplitude $t_{D}$ and $t_{I}$ are fixed in Ref. [36] by fitting the angular resolved photoemission spectroscopy experimental data; $H_{\mathrm{SOC}}$ is the atomic spin–orbit coupling of amplitude λ=0.01 eV [39], and $H_{\mathrm{TRI}}$ is the trigonal crystal field [36] of amplitude Δ=−0.005 eV [40]. Finally, $H_{v}$ parametrizes the effect of the confinement, which breaks the inversion symmetry and thus generates the so-called orbital Rashba [25], whose amplitude depends on the electric potential *v*. This term is responsible for the EE. In the region of low filling, a quadratic expansion in the quasimomentum $\vec{k}$ of the Hamiltonian leads to the effective Hamiltonian (the comparison between the exact microscopic model (Equation 1) and the effective Hamiltonian (Equation 2) breaks down for |k|>0.5 [37])
(2)$$H_{\mathrm{eff}}=\sum_{i=x,y,z}E_{i}\left(\vec{k}\right)\left(1-L_{i}^{2}\right)-\frac{\lambda }{2}\hat{L}\cdot \hat{S}-\frac{3\Delta }{2}L_{111}^{2}+F\left(\vec{k}\times \hat{L}\right)\cdot {\hat{n}}_{111}+\varepsilon_{0},$$
where $\vec{k}$ is expressed in units of the in-plane lattice constant $\tilde{a}=\sqrt{2/3}a_{0}=\sqrt{2/3}\cdot 0.3905$ nm, and $E_{i}$ is the renormalized dispersion expanded to second order at $\vec{k}$: (3)$$\begin{array}{cc} \; & E_{x}=0.13k_{X}^{2}-0.29k_{X}k_{Y}+0.29k_{Y}^{2}, \end{array}$$
(4)$$\begin{array}{c} E_{y}=0.13k_{X}^{2}+0.29k_{X}k_{Y}+0.29k_{Y}^{2}, \end{array}$$
(5)$$\begin{array}{c} E_{z}=0.37k_{X}^{2}+0.044k_{Y}^{2}, \end{array}$$
where 1 is the identity matrix, $L_{i}$ and $S_{i}$ are the *i*th components of the orbital and spin angular momentum operator for L=1 and S=1/2, $L_{111}$ is the projection of the angular momentum along the (111) direction, ${\hat{n}}_{111}$ is a unitary vector along the (111) direction, the term $\vec{k}\times \hat{L}$ is the orbital Rashba whose strength is included in the coefficient F=0.0035 eV (depending on *v*, which is fixed to 0.2 eV), and $\varepsilon_{0}$ is an energy constant. The expressions and the numerical values of the coefficients in Equation (Equation 2) can be found in Appendix A.

The combination of the atomic SOC and the orbital Rashba is translated into a generalized total angular momentum Rashba effect of the form $\hat{J}\times \vec{k}$, where $\vec{J}$ is the total angular momentum. The electronic band structure in the low energy region is shown in Figure 1. In the absence of SOC and the trigonal crystal field, all the bands would be degenerate in $\vec{k}=0$. The splitting between the doublets, due to these couplings, is smaller than in the most studied (001) LAO/STO interface, which is crucial for the results we find. The vicinity of the bands leads to a strong hybridization, which amplifies the spin and oribital EE. Near Γ, a linear Rashba splitting appears for the lowest Kramers doublet, whereas for the second doublet, a cubic splitting in $\vec{k}$ is found, differing from a simple description of a spin Rashba model [37]. Far away from Γ, the $d_{xy},d_{yz},d_{zx}$ character of the bands is restored. The region in which the crossover between these two behaviours occurs is the most sensible to the hybridization of the bands. By fixing the chemical potential to a benchmark value, we observe a non-trivial spin and orbital angular momentum texture on the Fermi surface in Figure 2. First, both the spin and the orbital angular momentum are wrapping around the Fermi contour. The orbital pattern shows that the in-plane component is higher when the Fermi surfaces of two doublets are close to one another, pointing in the same direction, which is a sign of hybridization. The textures for the other benchmark lines are found in Appendix B.

These textures are responsible for the spin and orbital EE when an electric field is included into the system. In linear response theory, the magnetization $m_{\alpha }$ along the α direction is
(6)$$m_{\alpha }=\chi_{\alpha \beta }E_{\beta },$$
where $\chi_{\alpha \beta }$ is the Edelstein susceptibility, and $E_{\beta }$ is the electric field in the β direction; $\chi_{\alpha \beta }$ is the sum of two contributions: a spin contribution $\chi_{\alpha \beta }^{S}$ and an orbital one $\chi_{\alpha \beta }^{L}$. We use the Boltzmann approach within the time relaxation approximation to compute the Edelstein susceptibility [22].

The magnetic moment per unit cell in the crystal is
(7)$$m_{\alpha }=\frac{\mu_{b}}{\hbar }S_{cell}\sum_{n}\int_{BZ}\frac{d^{2}\vec{k}}{{\left(2\pi \tilde{a}\right)}^{2}}\;\delta f\left(\vec{k}\right){\langle 2S_{\alpha }+L_{\alpha }\rangle }_{n}\left(\vec{k}\right)$$
where ${\langle S_{\alpha }\rangle }_{n}\left(\vec{k}\right)$ is the mean value over the eigenstates of the *n*th band, $\mu_{b}$ is the Bohr magneton, *ℏ* is the reduced Plank’s constant, $S_{cell}$ is the unit cell area, and δf is the modification of the thermal distribution $f_{\mathrm{th}}$ in the linear response regime, which is expressed as
(8)$$\delta f\left(\vec{k}\right)=-\tau_{0}q\tilde{a}\vec{E}\cdot \frac{\partial f_{\mathrm{th}}}{\partial \hbar \vec{k}}.$$

Here $\tau_{0}$ is the relaxation time, and *q* is the charge of the electrons. Therefore
(9)$$\chi_{\alpha \beta }^{O}=\left(-\frac{\tau_{0}q\mu_{b}}{\tilde{a}\hbar^{2}}S_{cell}\sum_{n}\int_{BZ}\frac{d^{2}\vec{k}}{{\left(2\pi \right)}^{2}}\;\frac{\partial f_{\mathrm{th}}}{\partial k_{\beta }}{\langle O_{\alpha }\rangle }_{n}\left(\vec{k}\right)\right),$$
where $O_{\alpha }=2S_{\alpha }$ or $L_{\alpha }$. Due to the anti-symmetric property of the $\chi_{\alpha \beta }$ [41], we need only to evaluate $\chi_{XY}$ (with $X=(\bar{1}10)$ and $Y=(\bar{1}\bar{1}2$) directions). The results are collected in Figure 3 both for the spin and the orbital susceptibility as a function of the chemical potential μ. We fixed the temperature to T=10 K and $\tau_{0}=3.4\times 10^{-12}$ s, the value of which is derived from the experimental mobility in Ref. [42]. Both susceptibilities behave non-monotonically and they are explicitly decomposed into the contributions of the three Kramers doublets in Equation (Equation 9), as also done in [22]. The spin susceptibility changes sign and presents a maximum and a minimum, suggesting that, in real systems, a magnetization reversal can be induced by appropriate gating (e.g., back gate control of the chemical potential).

In contrast, the orbital susceptibility is always of the order of $10^{-8}\mu_{B}$ mV${}^{-1}$, which is one order of magnitude greater than the spin susceptibility in the low energy region and above μ∼0.08 eV.

We demonstrate that a crucial ingredient for our results is the intermixing between different doublets. The reason is that the orbital Rashba term $\hat{L}\times \vec{k}$ induces an orbital angular momentum that is larger where the doublets are maximally hybridized. To demonstrate the role of the hybridization, we introduce $P^{\mu }=\mu \mu$ projector along the eigenstate μ of Hamiltonian (Equation 1) evaluated for $\vec{k}=0$. In this case, we identify three different states twice degenerate that we call Lσ (as low), Mσ (as middle), and Uσ (as up). We decompose the spin operator *S* as
(10)$$S_{\alpha }=\sum_{\mu \nu }P^{\mu }S_{\alpha }P^{\nu }=\sum_{\mu \nu }S_{\alpha }^{\mu \nu }.$$

A similar decomposition is adopted for *L*. By substituting Equation (Equation 10) in Equation (Equation 9), one can define an Edelstein susceptibility projected on the states for $\vec{k}=0$, $\chi_{\alpha \beta }^{O,\mu \nu }$, respecting the condition
(11)$$\chi_{\alpha \beta }^{O}=\sum_{\mu \nu }\chi_{\alpha \beta }^{O,\mu \nu }.$$

The magnitude of this quantity is an indicator of how much the hybridization of the doublets is important for EE or OEE. The values of $\chi_{XY}^{O,\mu \nu }$ for different benchmark chemical potential are represented in Figure 4. Thus we conclude that, for the first red peak of Figure 3a, there is a strong connection between the first two bands, indicating that this peak is described by a single doublet. However, there is the presence of hybridization with the second doublet as well, which is of the same order of magnitude of the intra-doublet interaction. The effect is even more evident for the angular momentum. By increasing the chemical potential, more doublets are filled, and the hybridization becomes more relevant. However, it is always true that the second intra-doublet contribution is zero, as seen from the 2×2 white square in Figure 4. This is a direct consequence of the absence of linear and quadratic splitting for the two bands in the second doublet. The intermediate doublet mediates the interaction between the first and the third doublet. This is confirmation of the relevance of the multiband model. Differently from the (001) interface, the (111) interface has the three doublets relatively close to one another, leading to this strong orbital hybridization.

## 3. Discussion

We have shown that the multiband character of the (111) LAO/STO is a key feature for the emerging non-linear spin and orbital EE. The strong SOC and the confining potential lead to a non-trivial Rashba interaction. Together with the orbital hybridization of the bands, this allows a spin and orbital magnetic moment in the presence of an in-plane external magnetic field. We have shown through a tight-binding model that the generalized Rashba effect can generate a non-linear spin EE that changes its sign with the chemical potential and can be modulated with an external gate. Moreover, the strong orbital character of the bands leads to an OEE, an order of magnitude higher than the spin effect at the very low and high fillings. Up to now, there is no direct evidence of orbital magnetization in the experiments. Because one can observe only the full magnetization, it is difficult to disentangle the spin from the orbital response [25]. However, by tuning to zero the spin Edelstein susceptibility, one can disentangle the two components, overcoming this problem. In Appendix C, we show how a $\vec{k}$-dependency on the scattering time changes the results. In principle, the strong orbital degeneracy and the hybridization of the bands could be enhanced by taking into account the contribution by impurities. However, direct computation within the framework of a *k*-dependent relaxation time shows small quantitative modification of results presented in the main text of this work. In particular, $\chi_{XY}^{S}$ vanishes at the same energy values predicted within the framework of a constant τ theory. Thus, our results provide a consistent picture of the system response. The proposal of tuning the spin response to zero, together with the new proposal of measuring the so-called orbital torque [43], makes the (111) LAO/STO interface suitable for investigating the orbital magnetization and represents a promising spin–orbitronic platform. The orbital angular momentum accumulated through the OEE can be transferred into an adjacent thin ferromagnet. The induced torque is sensitive to the interface crystallinity, and one can realize different experimental setups to capture the angular dependency on the torque. We remark that (111) KaTiO${}_{3}$-based heterostructures [28,44,45] have a similar crystalline structure with higher SOC, which could enhance the EE and OEE. Therefore, they could be an interesting system to further apply our analysis.

## Figures and Tables

**Figure 1 nanomaterials-12-02494-f001:**
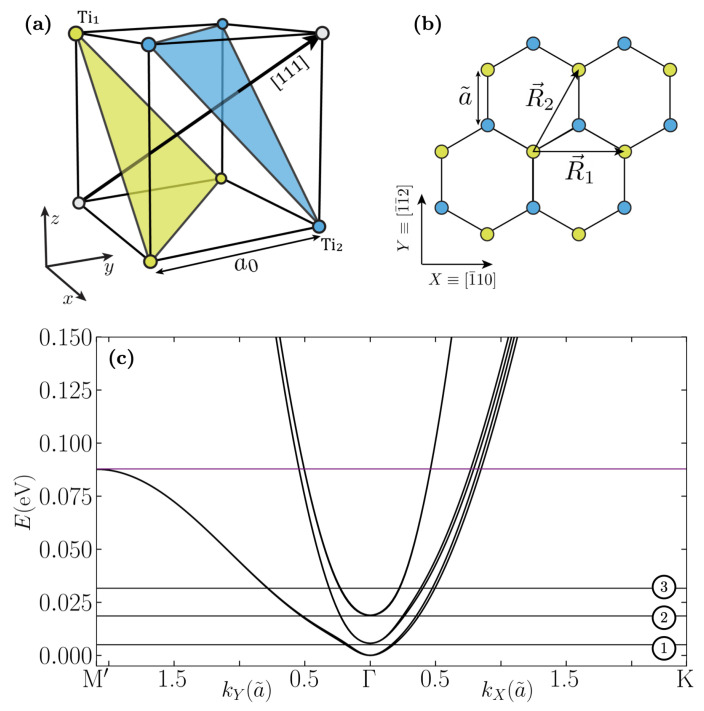
(**a**) Ti atoms in STO lattice, whose lattice constant is $a_{0}=0.3905$ nm. The blue and yellow dots represent atoms belonging to two non-equivalent planes. (**b**) Projection of the two non-equivalent planes of Ti over the (111) plane with our choice of primitive vectors ${\vec{R}}_{1}$ and ${\vec{R}}_{2}$ and $\tilde{a}=\sqrt{2/3}a_{0}$. (**c**) Band structure along two different directions in the Brillouin zone. The purple benchmark line corresponds to a Lifshitz transition (see Appendix C).

**Figure 2 nanomaterials-12-02494-f002:**
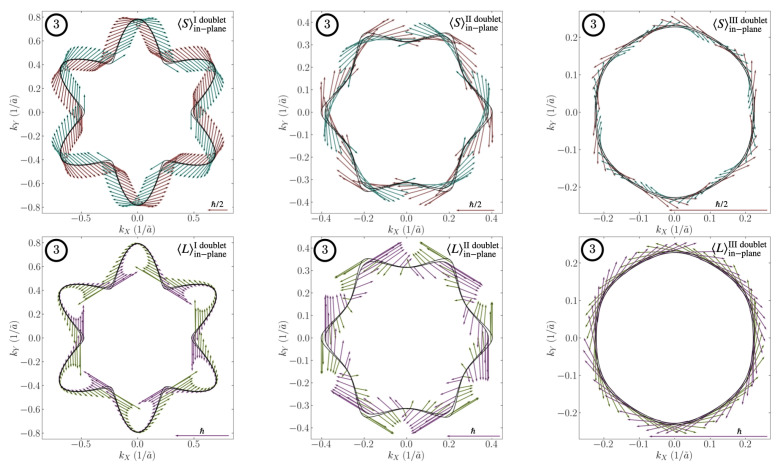
In-plane spin (**upper panel**) and orbital angular momentum (**lower panel**) textures for the three doublets with the chemical potential fixed to the value corresponding to the benchmark line 3 in Figure 1. The red and green arrows represent the mean value of the in-plane component of the operator for the external band, and the blue and purple refer to the internal component. The mean value of the generic operator *O* is evaluated as $\langle O\rangle =\sqrt{{\langle O_{\bar{1}10}\rangle }^{2}+{\langle O_{\bar{11}2}\rangle }^{2}}$.

**Figure 3 nanomaterials-12-02494-f003:**
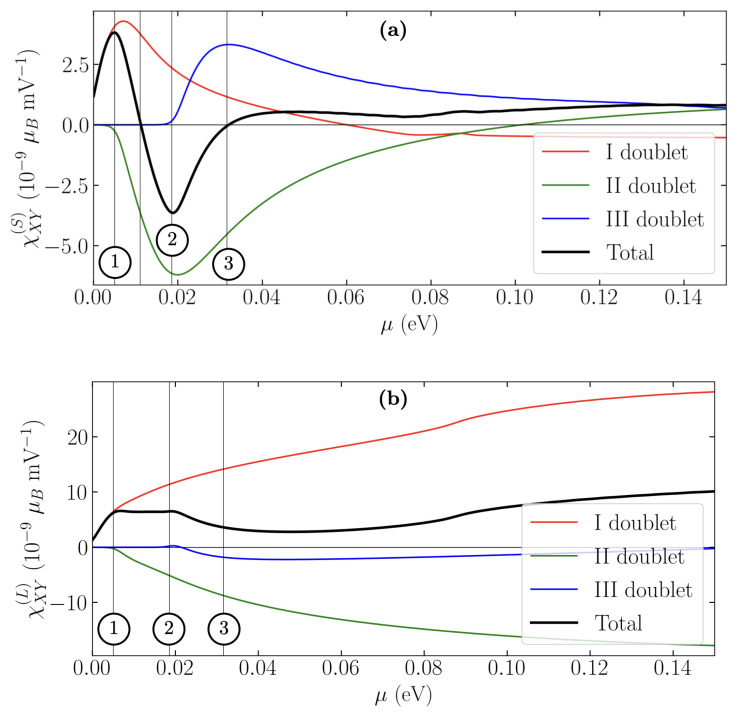
Spin (**a**) and orbital (**b**) Edelstein coefficient as a function of the chemical potential. The different colours correspond to the contribution of a specific Kramers doublet.

**Figure 4 nanomaterials-12-02494-f004:**
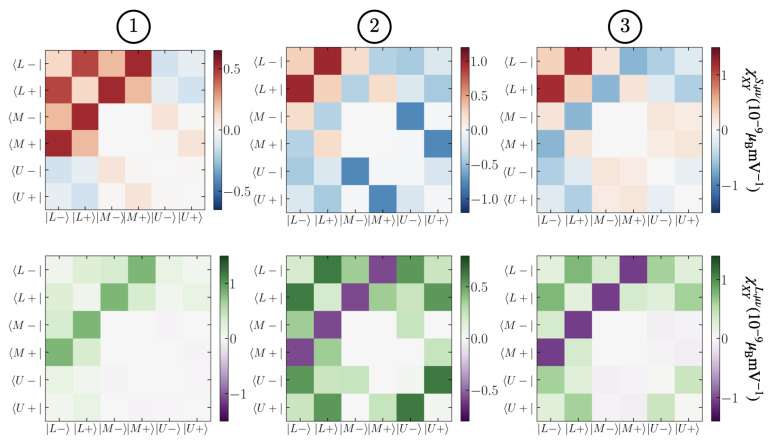
Spin (**upper panel**) and orbital (**lower panel**) Edelstein susceptibility projected over the *L*, *M*, and *U* states. The chemical potential μ is fixed at values 1, 2, and 3, referring to Figure 1.

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
