# Peer review of "Tunable Spin and Orbital Edelstein Effect at (111) LaAlO3/SrTiO3 Interface"

_nanomaterials, 2022, doi:10.3390/nano12142494_

Round 1

Reviewer 1 Report

The theoretical manuscript discusses tight-binding and Boltzmann equation transport calculations on the existence and interplay between the spin Edelstein effect and the orbital-magnetization Edelstein effect at the less-studied (111) interface of LaAlO3 and SrTiO3.  The manuscript is quite complete and its results and conclusions are credible within the models and parameter values used by the authors.  The manuscript will be of substantial and timely interest to the community, since both the oxide materials system and the orbital Edelstein effect are of increasing significance.  One conclusion that is particularly interesting is that the interplay and magnitude of the spin and orbital Edelstein effects have intricate origins and may depend on the details of hybridization in the band structure at the specific interface.  Even if specific values of the coefficients obtained in the manuscript may change depending on models, that insight would remain and will be useful for the community, especially for the design and interpretation of future experiments.  While the (111) interface has been studied in some detail in certain aspects and while the methods used in the manuscript don’t break new ground for oxide materials, the theoretical predictions are significant and new. That is due to the focus on the spin Edelstein effect and its gate dependence (which allows zeroing) and to the prediction of a strong orbital Edelstein effect.  The manuscript also contains sufficient detail in the main text and appendices to independently gauge the results if a computational reader wants to.  The manuscript is overall well written.  While the science deserves publication, some figures need revision for clarity.  In Figs. 2 and A1 the labels along the axes are nearly illegible, too small and in a faint color.  The arrows, while very useful, need a magnifying glass to make out.  The inset of Fig. A3a has similar problems.  Legends in these figures are too small also. 

Author Response

Response to Reviewer 1 Comments

We thank the Referee for the positive assessment of the manuscript. 

Point 1:  While the science deserves publication, some figures need revision for clarity.  In Figs. 2 and A1 the labels along the axes are nearly illegible, too small and in a faint color.  The arrows, while very useful, need a magnifying glass to make out.  The inset of Fig. A3a has similar problems.  Legends in these figures are too small also. 

Response 1: We changed all the figures mentioned by the referee to improve the visualization.

Reviewer 2 Report

In this work Trama an co-authors started from a tight-binding model approach to study the classical LAO/STO (111) interface, then calculated the spin and orbital Edelstein coefficient via linear response theory. They found that the multiband nature of the (111) LAO/STO is the key feature for the emerging spin and orbital Edelstein effect. Overall the work is well presented and I believe that the work can be accepted after some minor tweaks. I only have two suggestions: (1) Please explain how to experimentally distinguish the spin EE from the OEE. (2) It has already been known that the LAO/STO interface can be employed to generate spin currents or spin-orbit torques acting upon an adjacent ferromagnetic layer. Does this OEE can have an add-on effect?

Author Response

Response to Reviewer 2 Comments

We thank the referee for the positive assessment on our work.

Point 1: Please explain how to experimentally distinguish the spin EE from the OEE. It has already been known that the LAO/STO interface can be employed to generate spin currents or spin-orbit torques acting upon an adjacent ferromagnetic layer. Does this OEE can have an add-on effect?

Response 1: As far as we are aware, there is no experimental methodology to distinguish with certainty the two components of the magnetization induced by the EE and OEE. In fact, as seen from Eq.(7) in the paper, the observable magnetization is the sum of the spin and the orbital contributions. As we mention in the text, a possibility to disentangle the two components is given by choosing the chemical potential in order to tune to zero the spin magnetization. 
Another possibility is provided by performing a spin-torque and a orbital-torque experiment into a ferromagnet adjacent to the 2DEG. In fact in this case the EE and OEE generate respectively a spin accumulation and an orbital angular momentum accumulation, which result into different torques: the orbital angular momentum, in fact, has to be transferred into the ferromagnet by accounting for the interface crystallinity[1], which results into a dependence on the coupling between the interfaces. However, despite this, the orbital angular momentum is converted into a spin momentum due to the spin-orbit coupling into the ferromagnet, acting as an add-on contribution with respect to the spin one. A way to discriminate them is provided by preparing the samples in different experimental setups, to vary the interface cristallinity, and observing the changes in the produced torques. We include a sentence in the discussion in order to comment on this point.

We also want to thank again the referee for prompting us to deepen our understanding of these points, since we believe that it strongly improved the quality of the paper.

[1] Dongwook Go and Hyun-Woo Lee. Orbital torque: Torque generation by
orbital current injection. Phys. Rev. Research, 2:013177, Feb 2020.

Reviewer 3 Report

1. In (2), what is a symbol which looks like 1? What does it do with E_i?

2. The spin accumulation can be probed by an inverse spin Hall effect. It would be nice to add a sentence explaining if the OEE results in an orbital magnetization accumulation and if it could also be detected by the ISHE.

3. S and L enter in (7) on equal footing. Is it because L was substituted with J in (2)? Will still be the term \lambda/2 LS present? Does it contribute in (7)?  Why 2S and 1L?

4. (7) and Fig.4 imply that SEE and OEE are about the same. Why "an
orbital Edelstein effect is an order of magnitude larger then the spin one"? A combination of parameters/materials?

Author Response

Response to Reviewer 3 Comments

Point 1: In (2), what is a symbol which looks like 1? What does it do with $E_i$?

Response 1: In Eq(2) the symbol indicates the identity matrix and act multiplicatively on Ei. We include a sentence in the text in order to explain it.

Point 2: The spin accumulation can be probed by an inverse spin Hall effect. It would be nice to add a sentence explaining if the OEE results in an orbital magnetization accumulation and if it could also be detected by the ISHE.

Response 2: In principle the orbital magnetization, which is summed up to the spin one, generates a total magnetization. The inverse spin Hall effect would be sensitive to the total magnetization and so the OEE should play the role of an add-on term in the generation of a spin current.

Point 3: S and L enter in (7) on equal footing. Is it because L was substituted with J in (2)? Will still be the term $\lambda/2$ LS present? Does it contribute in (7)?  Why 2S and 1L?

Response 3: The Zeeman coupling in a crystal is of the form B⋅(g S+L), where the factor g is gyromagnetic factor. However, the magnetic energy is of the form m⋅B.  Therefore, in Eq(7) we identify the magnetization m∼(2S+L), where g=2 as the free electron gyromagnetic factor. Therefore, the total angular momentum J is not invoked in order to obtain that formula. About the atomic spin-orbit coupling, it is of course present in the Hamiltonian used to find the eigenstates over which we evaluate the mean values of S and L.

Point 4: (7) and Fig.4 imply that SEE and OEE are about the same. Why "an orbital Edelstein effect is an order of magnitude larger then the spin one"? A    combination of parameters/materials?

Response 4:  The results in Fig.4 show that the maximum of χS is 3 X 10-9 in unit of μB m V-1, and after the benchmark chemical potential point 3, almost saturates to 10-9; while χL, after the benchmark chemical potential 1, is higher than 5 X 10-9, reaching smaller values only in the depletion region barely between 0.03 eV and 0.08 eV. Our statement was not meant to hold throughout the whole chemical potential range, but as a general descriptions of the behaviour at large fillings. In the text we modify the abstract, the text (page 5) and the discussion in order to clarify the point.